# Kinetics of Anti-Hepatitis B Surface Antigen Titers in Nurse Students after a Two-Year Follow-Up

**DOI:** 10.3390/vaccines8030467

**Published:** 2020-08-21

**Authors:** Maria Gabriella Verso, Claudio Costantino, Alessandro Marrella, Palmira Immordino, Francesco Vitale, Emanuele Amodio

**Affiliations:** 1Occupational Health Unit, Department of Health Promotion Sciences, Maternal and Infantile Care, Internal Medicine and Medical Specialties “G. D’Alessandro”, University of Palermo, 90133 Palermo, Italy; 2Section of Hygiene, Department of Health Promotion Sciences, Maternal and Infantile Care, Internal Medicine and Medical Specialties “G. D’Alessandro”, University of Palermo, 90133 Palermo, Italy; claudio.costantino01@unipa.it (C.C.); alessandro.marrella@unipa.it (A.M.); palmira.immordino@gmail.com (P.I.); francesco.vitale@unipa.it (F.V.); emanuele.amodio@unipa.it (E.A.)

**Keywords:** hepatitis B virus (HBV), occupational visit, HCWs health, persistence of protection, vaccine booster, anamnestic response

## Abstract

Infection caused by hepatitis B virus (HBV) can be prevented through a safe and effective vaccine. This study analysed the kinetics of serum antibodies against hepatitis B surface antigen (HBsAg) (anti-HBs) titers in relation to previous vaccine boosters in Italian nursing students who were followed up for two years. Serum anti-HBs titers were evaluated at the first visit, after vaccine booster (if required) and at visit after two years. Overall, 483 students (mean age = 21.7 years; SD = 3.7) with median anti-HBs IgG titer of 6 mUI/mL (interquartile range (IQR) = 0–34) were enrolled. A total of 254 (52.5%) students with a titer lower than 10 mIU/mL were offered an anti-HBV booster at the first visit. Among these students, an exponential relation between anti-HBs IgG titer, one month after HBV booster and anti-HBs IgG titer two years later was found (*y* = 3.32 exp (0.0045x); R^2^ = 0.48; *p* < 0.001). Students with anti-HBV titer higher than 10 mIU/mL (*N* = 229) were followed up, and anti-HBs IgG titers at follow-up visit linearly correlated with anti-HBV baseline titers (*y* = 0.86x + 26.2; R^2^ = 0.67; *p* < 0.001). A decrease in anti-HBs titers can be expected a few years after the anti-HBV booster dose. This reduction is more pronounced than that observed in students not administered the booster dose and is exponential with respect to basal titers assessed after the booster dose.

## 1. Introduction

Hepatitis B virus (HBV) is a double-stranded DNA virus that belongs to the Hepadnavirus family. It has been classified into 10 genotypes (A–J) and more than 40 sub-genotypes [1,2].

Although HBV is a vaccine-preventable disease, it is still one of the most common infectious viral diseases worldwide. It has a different degree of endemicity, evaluated based on the seroprevalence of the hepatitis B surface antigen (HBsAg), and ranges from >8% (high endemicity) to 2–8% (intermediate endemicity) and <2% (low endemicity) [3]. In the European region, it has been estimated that about 1.6% of the population is infected [4].

HBV burden is not negligible; in fact, in 2015 approximately 257 million people were living with chronic HBV infection and 887,000 died worldwide because of the disease or its clinical complications [5].

The virus is transmitted by activities involving percutaneous or mucosal contact with blood and other body fluids of the infected person, potentially even by indirect contact, as the virus can remain viable outside the human body, even for several months [6,7].

Perinatal or vertical transmission to newborns is also possible and appears to be the most important factor in determining the high prevalence in endemic areas [8].

The natural history of the disease ranges from asymptomatic presentations to life-threatening illnesses such as acute and chronic hepatitis, cirrhosis, and hepatocellular carcinoma [9,10].

A vaccine against HBV has been available since 1981, and Italy was one of the first countries to introduce mandatory universal vaccination for newborns and 12-year-old adolescents in 1991 as well as adults belonging to high-risk groups [11]. The vaccination of adolescents was stopped in 2004, while the immunization of newborns was maintained, resulting in 24 cohorts of subjects being immunized against HBV in 12 years [11,12].

Healthcare workers (HCWs) are the group most at risk of infection and, for this reason, according to the Italian national vaccination plan, it is essential to vaccinate them before they start work activity [13].

According to these considerations, serum antibodies against HBsAg (anti-HBs) can be evaluated to check immunity in HCWs, including students who attend medical university courses [14,15]. A titer higher than 10 mIU/mL is generally considered a threshold to be protected against the infection, but a possible decline in the antibody levels has been observed [16,17].

However, there is a common agreement that a responder to the primary dose should be considered protected despite the actual antibody titer [6]. In the past several years, the international scientific community has raised a debate about the need for a booster dose [18].

It is clear that there is a need for more studies to increase the knowledge about HBV vaccination, including from an immunological point of view. To reduce this lack of knowledge, the main aim of the present study was to evaluate anti-HBs titers in a sample of university nursing students followed up for two years, assessing the role of anti-HBV booster in relation to the time to decay of anti-HBs titers.

## 2. Materials and Methods

An observational study was carried out by considering 637 students who, from November 2015 to April 2020, attended the nursing school at the University of Palermo, Italy. Students were examined during their first and third academic years in order to evaluate occupational risks. For each student, a standardized form was filled out with information regarding sociodemographic variables (age, gender, country of origin), personal anamnesis (relatives’ diseases and personal remote and proximate pathologies), and previous HBV vaccinations (at infancy or adolescence).

Moreover, a medical examination was performed, and a blood sample was obtained to evaluate serum HBsAg, anti-HBs, and anti-HBc. All students were negative for anti-HBc, and those who were negative for HBsAg and with anti-HBs titers lower than 10 mIU/mL were subsequently boosted and the titer was reassessed after one month. All enrolled nursing students were previously vaccinated against HBV, they were HBsAg negative, and none of them received a vaccine booster before the enrolment.

All students with a titer ≤10 mIU/mL at the first visit were boosted within one month, after they had produced the vaccination booklet. A minority of students who were not able to produce this documentation were asked to obtain a vaccine certification by the Local Health Unit and were vaccinated only after the acquisition of such certification.

Only students who met the following inclusion criteria were considered in the analyses:Those who attended both occupational visits in the first and third years.Those who underwent all laboratory analyses (in the first year, after HBV booster if required, and in the third year).

As 154 students did not meet the inclusion criteria, they were excluded from the analyses. Thus, analyses were finally carried out on a total of 483 (75.8% of the initial sample) students.

In accordance with Italian law, written informed consent was obtained from all subjects. The study was approved by the Ethics Committee of the University Hospital “P. Giaccone” of Palermo (Protocol number 26/2016, 19 October 2016).

### 2.1. Serological Tests

Serological analyses were performed using commercial chemiluminescence assays (VITROS anti-HBs assay on the VITROS ECI Immunodiagnostic System, Ortho-Clinical Diagnostics, High Wycombe, UK). The anti-HBs level was expressed as mIU/mL. Anti-HBs titers higher than 10 mIU/mL were considered as cut-off for statistical analyses, whereas those below 4.23 mIU/mL (the limit of detection of the assay) in the analyses were computed as 0 (undetectable).

### 2.2. Statistical Analysis

Absolute and relative frequencies were calculated for the categorical (qualitative) variables, and normally distributed quantitative variables were summarized by their means (standard deviations). The differences in the categorical variables were analysed using chi-squared tests (or Fisher’s exact test, when appropriate).

Means or medians were compared using either Student’s *t*-test or Wilcoxon rank sum test. To evaluate potential confounders due to sex and birth cohort, these last variables were included in two multivariate regression models, and subsequently excluded if not significantly associated with the level of anti-HBs.

The Akaike information criterion (AIC) was used to assess the fitting function of the different models. All information was entered into a database created with Excel 10.0 (Microsoft, Redmond, Seattle, WA, USA). All data were analysed using the R statistical software.

## 3. Results

Figure 1 describes the flow-chart diagram of the study. A total of 483 nursing students were considered in the analyses, 323 (66.9%) of whom were female (M:F ratio = 0.49). The mean age at first visit was 21.7 years (SD = 3.7), and all students were born between 1966 and 1999. As reported in Table 1, median anti-HBs IgG titer was 6 mIU/mL (interquartile range (IQR) = 0–34 mIU/mL), whereas the median was 41 mIU/mL (IQR = 18–123 mIU/mL) among 229 students with anti-HBs IgG titers above 10 mIU/mL. Statistically significantly higher levels were found in students vaccinated at adolescence (50 vs. 0 mIU/mL among those vaccinated at infancy, *p* < 0.001).

A total of 254 (52.5%) students had a titer below 10 mIU/mL and 185 of them (72.8%) had no detectable levels of antibodies. These students were boosted at the first occupational visit. In a multivariable logistic regression model, anti-HBV titers >1000 mIU/mL one month after HBV booster were found to be associated with higher anti-HBV basal titers (*p* < 0.001).

Among students requiring vaccine boosters, 22 (8.6%) were vaccinated during adolescence. In this group, vaccine boosting determined higher anti-HBV titers (median = 1000 mIU/mL; IQR 470–1000) than in those vaccinated at infancy (median = 554 mIU/mL; IQR = 108–1000), although this difference was not statistically significant (*p* = 0.08).

As shown in Figure 2, a statistically significant correlation was found between anti-HBs IgG titers one month after HBV booster and anti-HBs IgG titers two years later (at the third-year visit) among students administered boosters (function y = 3.32 exp (0.0045x); R^2^ = 0.48; *p* < 0.001).

To evaluate the potential confounding factors, sex and birth cohort were initially included in a multivariable regression analysis. However, these were subsequently excluded as they were not statistically significantly associated with anti-HBs IgG titers at the third-year visit and did not improve the overall fitting of the model.

Students with anti-HBV titer higher than 10 mIU/mL (*N* = 229) were only monitored during the two-year follow-up period. As reported in Figure 3, in these students anti-HBs IgG titers at two-year follow-up were linearly correlated with anti-HBV baseline titers (function y = 0.86x + 26.2; R^2^ = 0.67; *p* < 0.001).

Moreover, in this multivariable regression model, sex and birth cohort did not improve the general fitting of the model and were thus excluded from the analyses.

In Table 2, starting anti-HBs titers were compared with anti-HBs titers <10 mIU/mL at the two-year follow-up visit, and were stratified according to their anti-HBV booster status. A higher percentage of students who received the booster reported anti-HBs titers <10 mIU/mL after two years (18.1% vs. 14.4%), and this difference was statistically significant when considering only students with starting titers between 10 and 100 mIU/mL (55.7% vs. 17.9%; *p* < 0.001).

Finally, Figure 4 shows the modelled values obtained using the mathematical functions with the best fit between levels of anti-HBs titers, after two-year follow-up, in students who received and did not receive a booster vaccination. Students who had received a booster vaccination showed an exponential reduction in anti-HBs titers after the two-year follow-up, whereas those who did not receive the booster showed a linear reduction.

## 4. Discussion

Healthcare workers have the highest estimated infection rate for HBV [19]. Therefore, it is crucial that they obtain a proper occupational risk assessment before starting work activity to protect themselves and patients.

It is well documented that after primary HBV immunization, anti-HBs concentrations slowly decline and about 15–50% of vaccinated children will have low or undetectable concentrations of anti-HBs 5–15 years after vaccination [18].

At the same time, about 30–60% of adults vaccinated against HBV will have anti-HBs concentrations decline to <10 mIU/mL within 9–11 years [20].

Data obtained in the present study are consistent with these findings; more than half (52.5%) of the analysed students had titers <10 mIU/mL.

According to previously published investigations, this percentage was higher among students vaccinated at infancy who also had lower median anti-HBV titers [21].

Furthermore, in our study, the percentage of students with titers <10 mIU/mL was relatively higher than that found in other studies that evaluated medical students in Italy (40.8%, 29.3%, 12%, 15.8%) and lower with respect to that in other countries (72.4% in Malaysia, 59% in Israel) [22,23,24,25,26].

However, we can suppose that these percentages are strictly related to the greater number of years since the primary vaccination cycle (about two decades before) as well as the time of vaccination, as vaccination during infancy (86% of our students) is usually associated with a shorter immunity persistence [27,28].

When assessing healthcare students at the occupational visit, one major priority is to evaluate the effective immunization status after the initial HBV vaccination series. In the past several decades, several international guidelines have recommended that, after HBV vaccination, an anti-HBs serologic test result of >10 mIU/mL indicates immunity and, therefore no further routine doses or testing should be required [29,30].

In particular, these recommendations are consistent with the findings that, despite a decline in anti-HBs titers is expected over time, protection persisted, and vaccinated subjects with serologic evidence of HBV infection did not develop clinically significant sequelae [31,32,33,34,35]. It is noteworthy that about 38% of our students had no detectable anti-HBs titers and, thus, it can be assumed there was no response to the primary vaccination cycle. For this reason and consistent with the Centers for Disease Control and Prevention (CDC) recommendations, anti-HBs testing was performed for all students upon admission to medical school if a recent protective titer was not available. All students with anti-HBs titers <10 mIU/mL were thus administered a booster dose, and the large majority responded to the vaccination with a considerable increase in the anti-HBs titer after 1 month, as reported in nearly all other studies [24,36,37].

This finding suggests that a high proportion of vaccine recipients retain immune memory and would develop an anti-HBs response on exposure to HBV. As a second point of interest, we observed a good correlation between pre- and post-booster anti-HBs levels, contrary to a study carried out in Saudi Arabia, where the correlation was not significant [38].

Moreover, we analyzed the persistence of anti-HBs titers over time in boosted and not-boosted students. Some international studies have found that a major predictor of anti-HBs protective titers is the peak level of anti-HBs immediately achieved after primary immunization [39]. In other words, the higher the vaccine-induced anti-HBs concentration after the primary vaccination course, the longer the antibodies will persist. This finding is also probably true after booster doses. Moreover, our results confirm that these latter levels are well correlated with anti-HBs titers measured two years later. However, anti-HBs titers appear to have different kinetics in boosted and not-boosted subjects, with a rapid decay among the boosted subjects.

These different kinetics may not have a real impact from a clinical point of view, as it is expected that all subjects will be similarly protected against chronic active hepatitis. However, we believe that it is important to study these different expected anti-HBs titers for two main reasons.

First, clinicians need to know that boosted subjects can rapidly reduce their anti-HBs titers without the need for a further booster dose, especially in a shorter period. Second, very low anti-HBs titers could be protective against chronic active hepatitis but not against HBV infection. This last consideration may also have some public health implications in terms of spreading of the virus.

Finally, we have to highlight that our study unfortunately did not change the management of healthcare workers and, according to international guidelines, if they have anti-HBs titers <10 mmIU/mL and no evidence of a previous response to primary vaccination cycle, they have to be boosted. However, our findings change the knowledge about what occupational physicians can expect after boosting, as in a majority of boosted subjects, a very rapid decrement in anti-HBs titers can be observed after boosting.

## 5. Conclusions

The present study has some limitations. In particular, we analysed a relatively small sample size and this could be responsible for the low precision in the estimates and type II errors. We cannot exclude that we analysed some students who did not respond to the first vaccination cycle.

Moreover, a two-year follow-up could be a short period for assessing the long-term persistence of antibodies, especially considering that our sample could be affected by a selection bias due to the exclusive recruitment of young adults and without any evidence of abnormalities in the haemochrom. Thus, it is possible that the observed kinetics are not representative of and generalizable to the entire general population, including those with chronic health problems. Finally, for a small number of students, there was no evidence that they had completed the primary vaccination cycle and, although they had anti-HBs titers above 10 mIU/mL, it is not possible to exclude a role of this factor in our results.

Despite these limitations, our study suggests that an important decrease in anti-HBs titers can be expected just a few years after anti-HBV booster in a relatively high percentage of boosted subjects.

This reduction is quantitatively more pronounced than that observed in not-boosted students, and is exponential with respect to basal titers assessed after booster vaccination.

Further long-term follow-up studies will be required to explore in depth the relationship between HBV vaccination, booster, and duration of anti-HBV immunity. Moreover, evaluating the occurrence of anti-HBc in boosted students versus those with higher anti-HBs levels could help to assess if boosting increases the protection against HBV infection or the development of chronic infection.

Finally, it will be necessary to clarify how long immune memory persists and whether booster doses may still be needed in the long term to guarantee lifetime protection.

## Figures and Tables

**Figure 1 vaccines-08-00467-f001:**
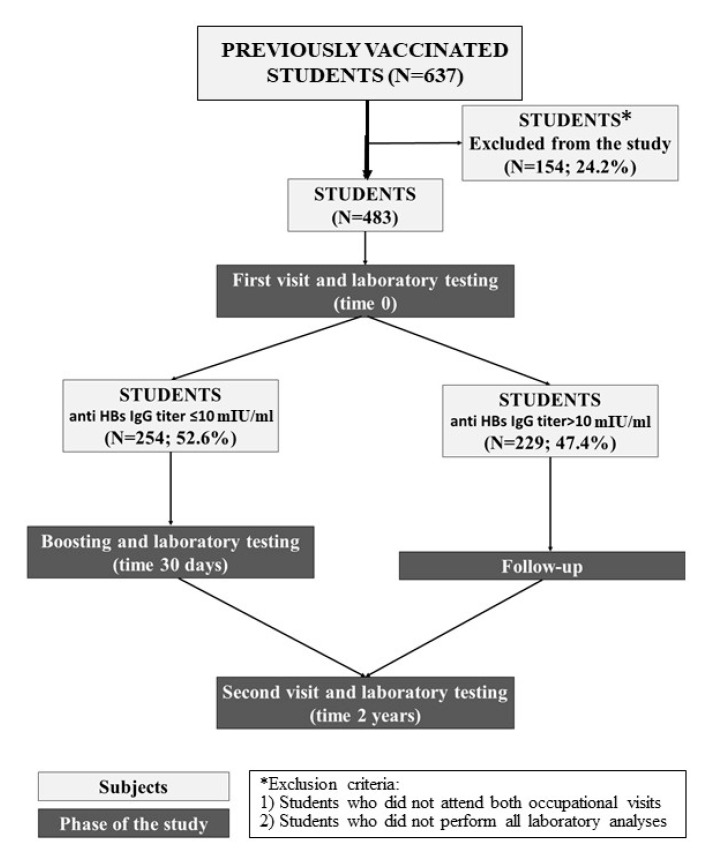
Flow-chart diagram according to the study phase (grey boxes: absolute frequencies of students; black boxes: phase of the study).

**Figure 2 vaccines-08-00467-f002:**
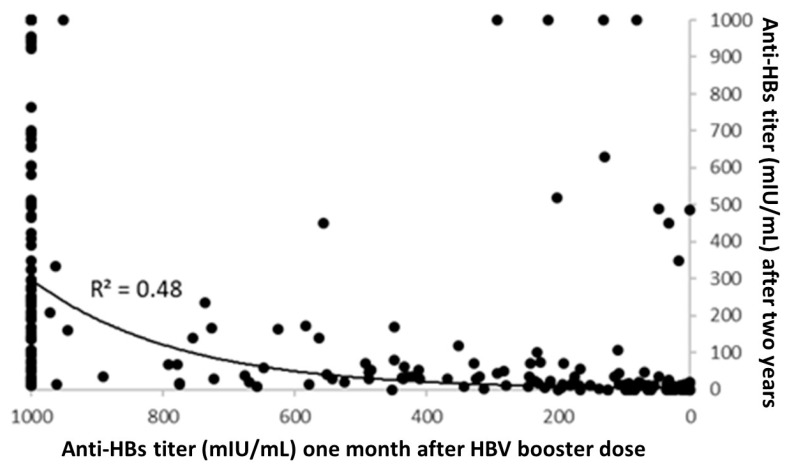
Correlation analysis between anti-HBs titers one month after hepatitis B virus (HBV) booster dose and anti-HBs titers two years later (R^2^ = 0.48; *p* < 0.001).

**Figure 3 vaccines-08-00467-f003:**
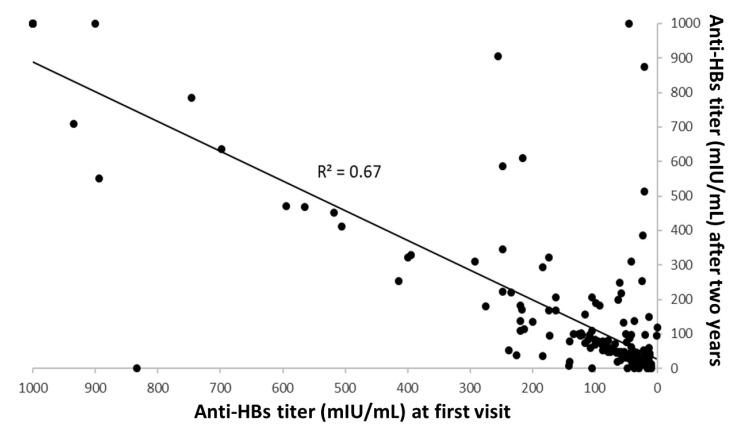
Correlation analysis between anti-HBs titers at first visit and at follow-up two years later (R^2^ = 0.67; *p* < 0.001).

**Figure 4 vaccines-08-00467-f004:**
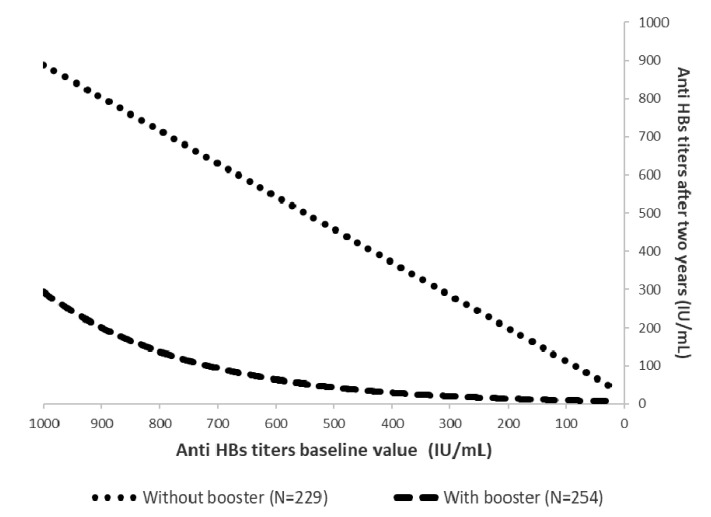
Modelled values of anti-HBs titers after two-year follow-up in students who received and those who did not receive a booster vaccination (values have been calculated according to the predictive function observed starting from an arbitrary value of 1000 mIU/mL).

**Table 1 vaccines-08-00467-t001:** General characteristics of nursing students in relation to the median of serum antibodies against hepatitis B surface antigen (anti-HBs) IgG titers at the first visit.

Variable	*N* (%)	Median Anti HBs IgG Titer at First Visit (mIU/mL) (IQR)
**Sex**		
Male	160 (33.1)	8 (0–43)
Female	323 (66.9)	0 (0–28)
**Birth cohort**		
1992 or less	101 (20.9)	38 (3–174)
1993–1994	84 (17.4)	0 (0–22)
1995–1996	198 (41.0)	0 (0–17)
>1997	100 (20.7)	6 (0–18)
**Vaccination timing**		
At adolescence	85 (17.6)	50 (8–219)
At infancy	398 (82.4)	0 (0–19)
**Booster after first visit**		
Yes	254 (52.6)	0 (0–0)
No	229 (47.4)	36 (16–105)
**Total**	483 (100)	6 (0–34)

**Table 2 vaccines-08-00467-t002:** Relative frequencies of students with hepatitis B surface antibody (anti-HBs) titers <10 mIU/mL at 2-year follow-up visit, stratified according to their basal value and anti-hepatitis B virus (HBV) boostering.

Starting Anti-HBs Titers Baseline * Value (mIU/mL)	HBs Titers <10 mIU/mL after Two Years		
	Boosted Students, *N* (%)	Not Boosted Students, *N* (%)	*p*-Value
0–100	34/61 (55.7%)	30/168 (17.9%)	<0.001
101–200	5/20 (25%)	2/24 (8.3%)	0.24
201–300	2/16 (12.5%)	0/15 (0%)	0.49
301–400	2/5 (40%)	0/2 (0%)	1
>400	3/149 (2%)	1/17 (5.9%)	0.34
Total	46/254 (18.1%)	33/229 (14.4%)	0.39

* For boosted students, the starting anti-HBs titer was considered after boosting.

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
