# Peer review of "Kinetics of Anti-Hepatitis B Surface Antigen Titers in Nurse Students after a Two-Year Follow-Up"

_vaccines, 2020, doi:10.3390/vaccines8030467_

Round 1

Reviewer 1 Report

The manuscript by Verso et al reports on a 2-year study of post-HBV vaccine boosting of nursing students with very low level of anti-HBs. Antibody titres 1 month and 2 years post boost have been monitored. There are several major issues that need attention and to a large extent require re-analysing the data.

Introduction

It should be indicated that 10mIU/ml is considered the limit of quantitation of anti-HBs and that the protective level is 100mIU/ml, not 10.

Results.

It is indicated that students were born between 1966 and 1999. Is this a typo for 1996? If not, as suggested by Table 1, reasons for vaccination prior to 1992 need to be given as well as the type of vaccine received.

The median anti-HBs titre of 6mIU/ml suggests quantification below the generally admitted cut-off of 10mIU/ml. Evidence for accurate quantification below that cut-off should be provided by the authors. If not, the stratification should simply be <10mIU/ml.

It is unclear whether or not any of the students carried anti-HBc. There are many reports showing a substantial percentage of people vaccinated at birth showing evidence of HBV infection by being anti-HBc positive. This information could be stratified between students with undetectable, <10 or >10mIU/ml anti-HBs.

For the large number of ‘vaccinated’ students with no detectable anti-HBS, what was the evidence for vaccination? Did all of them produce a vaccination booklet as evidence; if unavailable, how reliable vaccination was?

Figure 1 should be a composite showing results of students with undetectable anti-HBs (185) separate from those with positive but low anti-HBs (69). Figure 1 suggests that a relatively large number of students had very low or undetectable anti-HBs titre 1 month and 2 years after a single boost. This might suggest that some of them were not vaccinated or low responders.

In Figure 2, it is unclear what sample is taken as ‘first visit’: is it at baseline or is it at 1 month post-boost? If it is 1 month post-boost, correlation is expected since the decline of IgG level is universally according to similar kinetics.

The cohort of students included 398 vaccinated at birth and 85 during adolescence. The authors should examine and compare student response to boosting between these two groups for those with undetectable as well as <10mIU/ml. Such data might show a difference between boosting result 10 and 20 years after vaccination.

Discussion

The rapid decay of anti-HBs is similar to what is observed early after the peak of antibodies obtained after initial vaccination. It is likely that 10 or 20 years after boosting, an exponential kinetics will be observed when low levels are achieved. The difference observed is simply reflecting a shift in time.

The authors should examine the occurrence of anti-HBc two years after boosting for students with no or <10mIU/ml anti-HBs versus those with higher anti-HBs levels. This could be compared with results in not boosted students with low or undetectable anti-HBs. The really important issue not examined by the authors is whether or not boosting increases protection against HBV infection whether contact evidenced by anti-HBc or development of chronic infection. Longer-term monitoring of this limited cohort might provide such data.

Author Response

Answers to Reviewer #1

Reviewer #1:

Dear Reviewer,

Thank you for revising our manuscript and for your helpful comments and suggestions.

Below you will find a point-by-point answer to each your raised question. We hope this improved version of the manuscript could be considered suitable for publication on Vaccines.

Question: Introduction: It should be indicated that 10mIU/ml is considered the limit of quantitation of anti-HBs and that the protective level is 100mIU/ml, not 10.

Answer: We agree that serum levels above 100 mIU/ml should be considered as highly protective, however several Countries in the world use 10 mIU/ml as protection reference levels. In particular, in Italy, according to the European recommendations (Puro et al. Euro Surveill. 2005;10(10):pii=573), HCWs with postvaccinal anti-HBs levels, 1-2 months after vaccine completion, >10 mIU/mL are considered as responders and, thus, protected against HBV infection. For this reason, we would prefer to consider 10 mIU/ml as cut-off value. However, if required we would be happy to reconsider our position.

Q: Results. It is indicated that students were born between 1966 and 1999. Is this a typo for 1996? If not, as suggested by Table 1, reasons for vaccination prior to 1992 need to be given as well as the type of vaccine received.

A: We confirm that one student was born on 1966. However, this student has been vaccinated after 1992 with the same vaccine and the same standard three dose vaccination schedule used for adolescents.

Q: The median anti-HBs titre of 6mIU/ml suggests quantification below the generally admitted cut-off of 10mIU/ml. Evidence for accurate quantification below that cut-off should be provided by the authors. If not, the stratification should simply be <10mIU/ml.

A: The manufacturer indications report that limit of detection for anti-HBs titre is 4.23 mIU/mL and, thus, we used this value as minimum and the same cut-off is indicated by the CDC (https://wwwn.cdc.gov/nchs/data/nhanes/2007-2008/labmethods/hepb_s_e_met_hep_b_surface_antibody.pdf). For this reason, we have used 4.23 mIU/mL as minimum level and median anti-HBs titre observed in our students was 6 mIU/ml. This information has been added to the main text of the manuscript.

Q: It is unclear whether or not any of the students carried anti-HBc. There are many reports showing a substantial percentage of people vaccinated at birth showing evidence of HBV infection by being anti-HBc positive. This information could be stratified between students with undetectable, <10 or >10mIU/ml anti-HBs.

A: None of our students was positive to anti-HBc and thus we did not perform a stratification of the study sample.

Q: For the large number of ‘vaccinated’ students with no detectable anti-HBS, what was the evidence for vaccination? Did all of them produce a vaccination booklet as evidence; if unavailable, how reliable vaccination was?

A: All students requiring a booster dose have been requested for producing the vaccination booklet. A minority of students who were not able to produce this documentation were asked for obtaining a vaccine certification by the Local Health Unit and were vaccinated only after the acquisition of such certification. These statements have been added to the manuscript.

Q: Figure 1 should be a composite showing results of students with undetectable anti-HBs (185) separate from those with positive but low anti-HBs (69). Figure 1 suggests that a relatively large number of students had very low or undetectable anti-HBs titre 1 month and 2 years after a single boost. This might suggest that some of them were not vaccinated or low responders.

A: Since we have obtained vaccination certifications from all students, we are assuming that the entire cohort has been vaccinated. Otherwise, we cannot be sure that all of them have been responders to the vaccination and thus we added this consideration to the limitations of the study.

Q: In Figure 2, it is unclear what sample is taken as ‘first visit’: is it at baseline or is it at 1 month post-boost? If it is 1month post-boost, correlation is expected since the decline of IgG level is universally according to similar kinetics.

A: Figure 2 summarizes anti-HBs titers in 229 students who were not boosted. For this reason, IgG levels should be considered as at first visit (baseline time).

Q: The cohort of students included 398 vaccinated at birth and 85 during adolescence. The authors should examine and compare student response to boosting between these two groups for those with undetectable as well as <10mIU/ml. Such data might show a difference between boosting result 10 and 20 years after vaccination.

A: Among students requiring boosting, 22 (8.6%) were vaccinated during adolescence. In these subjects, vaccine boosting determined higher median anti-HBV titer (1,000 mIU/ml; IQR 470-1,000) than in those vaccinated at infancy (median=554; IQR=108-1,000) although this difference was not statistically significant (p=0.08). These results have been added to the results section of the manuscript. Moreover, it should be noted that in the multivariable regression analysis our results have been adjusted for the birth cohort (all students vaccinated during adolescence were born before 1992).

Q: The rapid decay of anti-HBs is similar to what is observed early after the peak of antibodies obtained after initial vaccination. It is likely that 10 or 20 years after boosting, an exponential kinetics will be observed when low levels are achieved. The difference observed is simply reflecting a shift in time.

A: We agree with you that probably the two different observed kinetics could be determined by a different time of observation. However, our key message, for occupational medicine and public health, is that after boosting a rapid exponential decline in anti-HBs could be observed. This is probably true also after a primary vaccination cycle, but we do not have availability of these last data and, thus, cannot confirm this possibility.

Q: The authors should examine the occurrence of anti-HBc two years after boosting for students with no or <10mIU/ml anti-HBs versus those with higher anti-HBs levels. This could be compared with results in not boosted students with low or undetectable anti-HBs. The really important issue not examined by the authors is whether or not boosting increases protection against HBV infection whether contact evidenced by anti-HBc or development of chronic infection. Longer-term monitoring of this limited cohort might provide such data.

A: Thank you very much for this suggestion. As reported, none of our students was found to be positive for anti-HBc. For this reason, a comparison between the two subgroups was not possible. However, we strongly agree that longer-term monitoring of this cohort might provide such data and increase general knowledge on this interesting topic. We added these considerations to the manuscript.

Reviewer 2 Report

The authors present an observational study of vaccinated nursing students, to evaluate the kinetics of anti-HBs over a 2 year follow-up, after a booster vaccination for non-responders, and without a boost for protected ones. They showed a more pronounced reduction of anti-HBs in boosted students, compared to non boosted ones.

The information is interesting, but I have several points to address: 1. 52,5% of students were non responders to previous vaccination, which is a very high proportion for young adults, for which I’d like to have some explanations !  2. Overall, presentation of the results lacks of some clarity and a flow chart is mandatory, as well as some initial results after the boost (see below); 3. I’d like to have more practical conclusions of the results obtained here: how to manage a health care worker, previously vaccinated, with a low anti-HBs titer, in light to this study ?

Introduction :

  • should be shortened
  • end of page 1: reference 6 doesn’t exactly refer to indirect contact as a route of HBV transmission, which I don’t believe in.

Material and methods:

  • previously vaccinated students were enrolled: how was the previous HBV vaccination certified / documented ? How many doses of vaccine did they previously receive ?
  • which vaccine, and which dose, as a booster, was administered ?
  • for more clarity, a flow chart of screened and enrolled subjects, should be provided.
  • Objectives of the study are not clearly presented.

Results:

  • first paragraph of page 4: results of anti-HBs response 1 month after the boost, should be presented; was there any other correlation in the response to the boost (sex for example, or age ?)
  • the late response results could also be presented …
  • line 131: students with anti-HBV titers higher than 10 mIU/ml (229): is this before or after the booster injection ? a flow chart would greatly improve overall comprehension !
  • in figure 2, baseline appears to be the first visit, before the boost, while in table 2, baseline was considered after the boost; this induces some confusion …
  • figure 3: it seems there is an inversion of the 2 lines, compared to the text !

Author Response

Answers to Reviewer #2

Dear Reviewer,

Thank you for revising our manuscript and for your helpful comments and suggestions.

Below you will find a point-by-point answer to each your raised question.

Question: The authors present an observational study of vaccinated nursing students, to evaluate the kinetics of anti-HBs over a 2 year follow-up, after a booster vaccination for non-responders, and without a boost for protected ones. They showed a more pronounced reduction of anti-HBs in boosted students, compared to non boosted ones. The information is interesting, but I have several points to address: 1. 52,5% of students were non responders to previous vaccination, which is a very high proportion for young adults, for which I’d like to have some explanations ! 

Answer: We apologize for this misunderstanding. In our study 254 students had anti-HBs titers <10 mmIU/ml and they were, thus, vaccinated. Probably, the vast majority of them were responders to the primary vaccination cycle but, with time, their IgG levels declined becoming too low for ensuring effective protection. Unfortunately, we do not have information about true not responders and, thus, in the manuscript we stated that “Finally, we cannot exclude that we have included in our analyses some students who were not responders to the first vaccination cycle”.

Q: Overall, presentation of the results lacks of some clarity and a flow chart is mandatory, as well as some initial results after the boost (see below).

A: As suggested, a flow-chart has been added.

Q: I’d like to have more practical conclusions of the results obtained here: how to manage a health care worker, previously vaccinated, with a low anti-HBs titer, in light to this study?

A: Thank you for the opportunity to clarify this point. Unfortunately, our study does not change the management of healthcare workers and, according to international guidelines, if they have anti-HBs titers <10 mmIU/ml and they do not have evidence for a previous response to primary vaccination cycle they have to be boosted. However, our findings change the knowledge about what occupational physicians could expect after boosting. In this sense, our results simply suggest that in a majority of boosted subjects, a very rapid decrement of anti-HBs titers could be observed after boosting. We added these considerations to the manuscript.

Q: Introduction: should be shortened. End of page 1: reference 6 doesn’t exactly refer to indirect contact as a route of HBV transmission, which I don’t believe in.

A: We shortened the introduction and evidences for HBV indirect transmission have been referred to a new reference (Williams et al. Clin Infect Dis. 2004 Jun 1; 38(11):1592-8).

Q: Material and methods: previously vaccinated students were enrolled: how was the previous HBV vaccination certified / documented ? How many doses of vaccine did they previously receive? which vaccine, and which dose, as a booster, was administered? for more clarity, a flow chart of screened and enrolled subjects, should be provided.

A: As requested from Reviewer #1, we have clarified that “All students with a titer ≤10 mIU/mL at the first visit were boosted within one month and after they have produced the vaccination booklet. A minority of students who were not able to produce this documentation were asked for obtaining a vaccine certification by the local health Unit and were vaccinated only after the acquisition of such certification”. Unfortunately, we did not record information about previous HBV primary cycle (how many doses, which dose etc and thus further analysed would be not possible. Otherwise, we added  a flow chart of screened and enrolled subjects (Figure 1).

Q: Objectives of the study are not clearly presented.

A: We apologize for this lack of clarity. We have thereforerephrased objectives of the study, as kindly requested.

Q: Results: first paragraph of page 4: results of anti-HBs response 1 month after the boost, should be presented; was there any other correlation in the response to the boost (sex for example, or age ?) the late response results could also be presented …

A: As requested by the Reviewer #1, we evaluated the role of time to vaccination and added the following sentence: “Among students requiring boosting, 22 (8.6%) were vaccinated during adolescence. In these subjects, vaccine boosting determined higher anti-HBV titers (median=1,000 mIU/ml; IQR 470-1,000) than in those vaccinated at infancy (median=554 mIU/ml; IQR=108-1,000) although this difference was not statistically significant (p=0.08)”. However, we would prefer to avoid an analysis on factors that are involved in response to booster since we have already published similar results elsewhere (see at Verso et al. Vaccines 2020, 8(1), 1; https://doi.org/10.3390/vaccines8010001). Otherwise, if required we will reconsider our purpose.

Q: line 131: students with anti-HBV titers higher than 10 mIU/ml (229): is this before or after the booster injection ? a flow chart would greatly improve overall comprehension !

A: The sentence is related to students before booster injections. As suggested, we have added a flow-chart for improving comprehension.

Q: in figure 2, baseline appears to be the first visit, before the boost, while in table 2, baseline was considered after the boost; this induces some confusion …

A: We agree with you and we used the term “baseline” only for first visit.

Q: figure 3: it seems there is an inversion of the 2 lines, compared to the text

A: We apologize for this mistake. Dotted lines have been modified according to our real findings.

Round 2

Reviewer 1 Report

The authors took into consideration most comments from this reviewer. However, one comment (below) was not sufficiently addressed, decreasing significantly the understanding and significance of the data.

The median anti-HBs titre of 6mIU/ml suggests quantification below the generally admitted cut-off of 10mIU/ml. Evidence for accurate quantification below that cut-off should be provided by the authors. If not, the stratification should simply be <10mIU/ml.

It is unclear whether or not any of the students carried anti-HBc. There are many reports showing a substantial percentage of people vaccinated at birth showing evidence of HBV infection by being anti-HBc positive. This information could be stratified between students with undetectable, <10 or >10mIU/ml anti-HBs.

The authors did not take proper care of this comment. They only provided the additional information that the anti-HBs LOD was 4.3mIU/ml. They should therefore stratify their cohort into undetectable anti-HBs (<4.3mIU/ml), low anti-HBs 4.3-10mIU/ml and anti-HBs positive. In Figure 2, a different colour code should be applied so that each group can be identified. The numbers summarising the median anti-HBs should be recalculated excluding samples below 4.3mIU/ml of anti-HBs.

Author Response

Dear Reviewer,

Thank you again for this further step in the reviewing our manuscript.

Below you will find a point-by-point answer to each your raised question. We hope this improved version of the manuscript could be considered suitable for publication on Vaccines.

Question: The authors took into consideration most comments from this reviewer. However, one comment (below) was not sufficiently addressed, decreasing significantly the understanding and significance of the data.

The median anti-HBs titre of 6mIU/ml suggests quantification below the generally admitted cut-off of 10mIU/ml. Evidence for accurate quantification below that cut-off should be provided by the authors. If not, the stratification should simply be <10mIU/ml.

Answer: As suggested in the manuscript we reported median anti-HBs titres for subjects with levels above 10 mIU/ml (page 4, line 125). Moreover, in the Material and Methods (page 3 line 104) we clarified that titers below 4.23 mIU/mL (the limit of detection of the assay) were computed as 0 (undetectable). All the analyses have been recalculated accordingly. On the other hands, we did not stratify our analyses into three further groups (anti-HBs titers 0 to 4.23, >4.23 to 10 and > 10 10 mIU/ml) since we are confident that the use of non-parametric statistical analyses (including the use of medians as measure of central tendency) should be not affected by the precise quantification of levels below 4.23 mIU/ml and that subjects with levels between >0 and 4.23 were only 35, thus being a very small proportion.

Finally, it should be noted that the analyses of anti-HBs titre <10 mIU/ml, although with the limitations that you have rightly highlighted, is quite frequent among researchers and thus the international literature has already evaluated this possibility as remarkable (for more information see at Yu-Sheng et al (2016), Pediatrics & Neonatology or Shunshun et al (2016), Hum Vac & Imm or Ganczak et al. BMC Infectious Diseases (2017) 17:515).

Question: It is unclear whether or not any of the students carried anti-HBc. There are many reports showing a substantial percentage of people vaccinated at birth showing evidence of HBV infection by being anti-HBc positive. This information could be stratified between students with undetectable, <10 or >10mIU/ml anti-HBs.

Answer: In the previous revision we have clarified that all students were negative for anti-HBc and, thus, a stratification for this variable would be non sense. In the manuscript we have reported that “All students resulted negative for anti-HBc” (page 3 line 80).

Question: The authors did not take proper care of this comment. They only provided the additional information that the anti-HBs LOD was 4.3mIU/ml. They should therefore stratify their cohort into undetectable anti-HBs (<4.3mIU/ml), low anti-HBs 4.3-10mIU/ml and anti-HBs positive. In Figure 2, a different colour code should be applied so that each group can be identified. The numbers summarising the median anti-HBs should be recalculated excluding samples below 4.3mIU/ml of anti-HBs. Dotted lines have been modified according to our real findings.

Answer: As discussed at point 1, we have updated analyses (anti HBs titers <4.23 mIU/ml have been considered as undetectable and thus computed as 0) but we would prefer to avoid a further stratification in three subgroups since we would have six final subgroups (boosted and not boosted stratified by 0-4.23; >4.23 to 10, and >10 mIU/ml, respectively). More subgroups would make the tables more confused and the small sample size of each subgroup would affect the power of the performed statistical tests. Moreover, all students included in the models had antiHBs titers >10 mIU/ml (for 254 students this value was obtained after boosting) and thus stratification would not improve the general quality of our results. For this reason we would like to maintain the current two subgroups (boosted and not boosted) without further stratification. However, we could reconsider this position if requested from the Editor.

Reviewer 2 Report

The authors responded to the majority of comments, and corrected the manuscript in accordance. I nevertheless have a few minor comments:

  • thanks for providing a helpful flow chart, in which I’d specify “previously HBV vaccinated students” in the first box
  • I’d also specify the reasons for non inclusion (154 students)
  • So, students had to provide a vaccination booklet, or a certification, to prove previous HBV immunization, this is a very important clarification; however, I strongly regret the absence of information of this previous vaccine schedule: was it complete in most of the case, or not ? If it is often incomplete (vaccine schedule in infancy or adolescence), it could explain the low rate of protected students at enrolment in this study …As the vaccine booklet has been showed, why wasn’t this not recorded ? It is a limitation of interpretation, which should be added in the discussion.

Author Response

Dear Reviewer,

Thank you for this further step in reviewing our manuscript and for your helpful comments and suggestions.

Below you will find a point-by-point answer to each your raised question.

Question: The authors responded to the majority of comments, and corrected the manuscript in accordance. I nevertheless have a few minor comments: thanks for providing a helpful flow chart, in which I’d specify “previously HBV vaccinated students” in the first box. I’d also specify the reasons for non inclusion (154 students).

Answer: The flow-chart has been modified according to your suggestions.

Question: So, students had to provide a vaccination booklet, or a certification, to prove previous HBV immunization, this is a very important clarification; however, I strongly regret the absence of information of this previous vaccine schedule: was it complete in most of the case, or not ? If it is often incomplete (vaccine schedule in infancy or adolescence), it could explain the low rate of protected students at enrolment in this study …As the vaccine booklet has been showed, why wasn’t this not recorded ? It is a limitation of interpretation, which should be added in the discussion.

Answer: We apologize for this limitation of the study. Unfortunately, as previously reported, we did not record these information and, thus, it is for us impossible to analyse them. Otherwise, since we have performed all the occupational visits, we are sure that only a minority of students (indicatively less than 10) did not complete the primary vaccination cycle and all of them have anti-HBs titers above 10 mIU/ml (this was the reason for which we did not add any note to the anamnestic schedule). We have reported these considerations among the limitations of the study (page 9 line 266).

Round 3

Reviewer 1 Report

The authors still did not address the comment below:

'It is unclear whether or not any of the students carried anti-HBc. There are many reports showing a substantial percentage of people vaccinated at birth showing evidence of HBV infection by being anti-HBc positive. This information could be stratified between students with undetectable, <10 or >10mIU/ml anti-HBs.'

This should be easily addressed since M&M indicate that nit-HBchad been tested. Examining this issue would provide important information regarding the efficacy or actuality of HBV vaccination.
